# Explanations based on the Missing: Towards Contrastive Explanations with Pertinent Negatives

**Amit Dhurandhar**[*]
IBM Research
Yorktown Heights, NY 10598
adhuran@us.ibm.com

**Pin-Yu Chen**[*]
IBM Research
Yorktown Heights, NY 10598
pin-yu.chen@ibm.com

**Ronny Luss**
IBM Research
Yorktown Heights, NY 10598
rluss@us.ibm.com

**Chun-Chen Tu**
University of Michigan
Ann Arbor, MI 48109
timtu@umich.edu

**Paishun Ting**
University of Michigan
Ann Arbor, MI 48109
paishun@umich.edu

**Karthikeyan Shanmugam**
IBM Research
Yorktown Heights, NY 10598
karthikeyan.shanmugam2@ibm.com

**Payel Das**
IBM Research
Yorktown Heights, NY 10598
daspa@us.ibm.com

## Abstract

In this paper we propose a novel method that provides contrastive explanations justifying the classification of an input by a black box classifier such as a deep neural network. Given an input we find what should be minimally and sufficiently present (viz. important object pixels in an image) to justify its classification and analogously what should be minimally and necessarily *absent* (viz. certain background pixels). We argue that such explanations are natural for humans and are used commonly in domains such as health care and criminology. What is minimally but critically *absent* is an important part of an explanation, which to the best of our knowledge, has not been explicitly identified by current explanation methods that explain predictions of neural networks. We validate our approach on three real datasets obtained from diverse domains; namely, a handwritten digits dataset MNIST, a large procurement fraud dataset and a brain activity strength dataset. In all three cases, we witness the power of our approach in generating precise explanations that are also easy for human experts to understand and evaluate.

## 1  Introduction

*Steve is the tall guy with long hair who does not wear glasses.* Explanations as such are used frequently by people to identify other people or items of interest. We see in this case that characteristics such as being tall and having long hair help describe the person, although incompletely. The absence of glasses is important to complete the identification and help distinguish him from, for instance, Bob who is tall, has long hair and wears glasses. It is common for us humans to state such contrastive facts when we want to accurately explain something. These contrastive facts are by no means a list of all possible characteristics that should be absent in an input to distinguish it from all other classes that it does not belong to, but rather a minimal set of characteristics/features that help distinguish it from the "closest" class that it does not belong to.

---

[*]First two authors have equal contribution.

In this paper we want to generate such explanations for neural networks, in which, besides highlighting what is minimally sufficient (e.g. tall and long hair) in an input to justify its classification, we also want to identify contrastive characteristics or features that should be minimally and critically *absent* (e.g. glasses), so as to maintain the current classification and to distinguish it from another input that is "closest" to it but would be classified differently (e.g. Bob). We thus want to generate explanations of the form, "*An input $x$ is classified in class $y$ because features $f_i, \cdots , f_k$ are present and because features $f_m, \cdots , f_p$ are absent*." The need for such an aspect as what constitutes a good explanation has been stressed on recently [12]. It may seem that such crisp explanations are only possible for binary data. However, they are also applicable to continuous data with *no explicit discretization or binarization required*. For example, in Figure 1, where we see hand-written digits from MNIST [40] dataset, the black background represents no signal or absence of those specific features, which in this case are pixels with a value of zero. Any non-zero value then would indicate the presence of those features/pixels. This idea also applies to colored images where the most prominent pixel value (say median/mode of all pixel values) can be considered as no signal and moving away from this value can be considered as adding signal. One may also argue that there is some information loss in our form of explanation, however we believe that such explanations are lucid and easily understandable by humans who can always further delve into the details of our generated explanations such as the precise feature values, which are readily available. Moreover, the need for such simple, clear explanations over unnecessarily complex and detailed ones is emphasized in the recent General Data Protection Regulation (GDPR) passed in Europe [41].

In fact, there is another strong motivation to have such form of explanations due to their presence in certain human-critical domains. In medicine and criminology there is the notion of pertinent positives and pertinent negatives [15], which together constitute a complete explanation. *A pertinent positive (PP) is a factor whose presence is minimally sufficient in justifying the final classification*. On the other hand, *a pertinent negative (PN) is a factor whose absence is necessary in asserting the final classification*. For example in medicine, a patient showing symptoms of cough, cold and fever, but no sputum or chills, will most likely be diagnosed as having flu rather than having pneumonia. Cough, cold and fever could imply both flu or pneumonia,

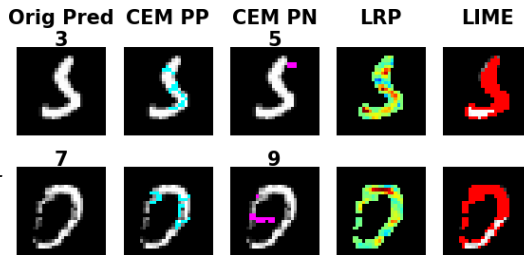

Figure 1: CEM versus LRP and LIME on MNIST. PP/PN are highlighted in cyan/pink respectively. For LRP, green is neutral, red/yellow is positive relevance, and blue is negative relevance. For LIME, red is positive relevance and white is neutral.

however, the absence of sputum and chills leads to the diagnosis of flu. Thus, sputum and chills are pertinent negatives, which along with the pertinent positives are critical and in some sense sufficient for an accurate diagnosis.

We thus propose an explanation method called *contrastive explanations method* (CEM) for neural networks that highlights not only the pertinent positives but also the pertinent negatives. This is seen in Figure 1 where our explanation of the image being predicted as a 3 in the first row does not only highlight the important pixels (which look like a 3) that should be present for it to be classified as a 3, but also highlights a small horizontal line (the pertinent negative) at the top whose presence would change the classification of the image to a 5 and thus should be absent for the classification to remain a 3. Therefore, our explanation for the digit in row 1 of Figure 1 to be a 3 would be: *The row 1 digit is a 3 because the cyan pixels (shown in column 2) are present and the pink pixels (shown in column 3) are absent*. This second part is critical for an accurate classification and is *not* highlighted by any of the other state-of-the-art interpretability methods such as layerwise relevance propagation (LRP) [1] or locally interpretable model-agnostic explanations (LIME) [30], for which the respective results are shown in columns 4 and 5 of Figure 1. Moreover, given the original image, our pertinent positives highlight what should be present that is necessary and sufficient for the example to be classified as a 3. This is not the case for the other methods, which essentially highlight positively or negatively relevant pixels that may not be necessary or sufficient to justify the classification.

**Pertinent Negatives vs Negatively Relevant Features:** Another important thing to note here is the conceptual distinction between pertinent negatives that we identify and negatively correlated or relevant features that other methods highlight. The question we are trying to answer is: *why is*

*input x classified in class y?*. Ergo, any human asking this question wants all the evidence in support of the hypothesis of $x$ being classified as class $y$. Our pertinent positives as well as negatives are evidences in support of this hypothesis. However, unlike the positively relevant features highlighted by other methods that are also evidence supporting this hypothesis, the negatively relevant features by definition do not. Hence, another motivation for our work is that we believe when a human asks the above question, they are more interested in evidence supporting the hypothesis rather than information that devalues it. This latter information is definitely interesting, but is of secondary importance when it comes to understanding the human's intent behind the question.

Given an input and its classification by a neural network, CEM creates explanations for it as follows:

(1) It finds a minimal amount of (viz. object/non-background) features in the input that are sufficient in themselves to yield the same classification (i.e. PPs).

(2) It also finds a minimal amount of features that should be *absent* (i.e. remain background) in the input to prevent the classification result from changing (i.e. PNs).

(3) It does (1) and (2) "close" to the data manifold using a state-of-the-art convolutional autoencoder (CAE) [25] so as to obtain more "realistic" explanations.

We enhance our methods to do (3), so that the resulting explanations are more likely to be close to the true data manifold and thus match human intuition rather than arbitrary perturbations that may change the classification. Of course, learning a good representation using an autoencoder may not be possible in all situations due to limitations such as insufficient data or bad data quality. It also may not be necessary if all combinations of feature values have semantics in the domain or the data does not lie on low dimensional manifold as is the case with images.

We validate our approaches on three real-world datasets. The first is MNIST [40], from which we generate explanations with and without an autoencoder. The second is a procurement fraud dataset [9] from a large corporation containing millions of invoices that have different risk levels. The third one is a brain functional MRI (fMRI) imaging dataset from the publicly accessible Autism Brain Imaging Data Exchange (ABIDE) I database [11], which comprises of resting-state fMRI acquisitions of subjects diagnosed with autism spectrum disorder (ASD) and neurotypical individuals. For the latter two cases, we do not consider using autoencoders. This is because the fMRI dataset is insufficiently large especially given its high-dimensionality. For the procurement data, all combination of allowed feature values are (intuitively) reasonable. In all three cases, we witness the power of our approach in creating more precise explanations that also match human judgment.

## 2   Related Work

Researchers have put great efforts in devising algorithms for interpretable modeling. Examples include establishment for rule/decision lists [39, 36], prototype exploration [19, 13], developing methods inspired by psychometrics [17] and learning human-consumable models [6]. Moreover, there is also some interesting work which tries to formalize and quantify interpretability [10].

A recent survey [24] looks primarily at two methods for understanding neural networks: a) Methods [26, 27] that produce a prototype for a given class, b) Explaining a neural network's decision on an input by highlighting relevant parts [1, 20, 30, 33]. Other works also investigate methods of the type (b) for vision [34, 35, 29] and NLP applications [22]. Most of the these explanation methods, however, focus on features that are present, even if they may highlight negatively contributing features to the final classification. As such, they do not identify features that should be necessarily and sufficiently present or absent to justify for an individual example its classification by the model. There are methods which perturb the input and remove features [32], however these are more from an evaluation standpoint where a given explanation is quantitatively evaluated based on such procedures.

Recently, there has been a piece of work [31] that tries to find sufficient conditions to justify classification decisions. As such, this work tries to find feature values whose presence conclusively implies a class. Hence, these are global rules (called anchors) that are sufficient in predicting a class. Our PPs and PNs on the other hand are customized for each input. Moreover, a dataset may not always possess such anchors, although one can almost always find PPs and PNs. There is also work [43] that tries to find stable insight that can be conveyed to the user in a (asymmetric) binary setting for smallish neural networks.

It is also important to note that our method is related to methods that generate adversarial examples [5, 7]. However, there are certain key differences. Firstly, the (untargeted) attack methods are largely unconstrained where additions and deletions are performed simultaneously, while in our case for PPs and PNs we only allow deletions and additions respectively. Secondly, our optimization objective for PPs is itself distinct as we are searching for features that are minimally sufficient in themselves to maintain the original classification. As such, our work demonstrates how attack methods can be adapted to create effective explanation methods.

## 3   Contrastive Explanations Method

This section details the proposed contrastive explanations method. Let $\mathcal{X}$ denote the feasible data space and let $(\mathbf{x}_0, t_0)$ denote an example $\mathbf{x}_0 \in \mathcal{X}$ and its inferred class label $t_0$ obtained from a neural network model. The modified example $\mathbf{x} \in \mathcal{X}$ based on $\mathbf{x}_0$ is defined as $\mathbf{x} = \mathbf{x}_0 + \boldsymbol{\delta}$, where $\boldsymbol{\delta}$ is a perturbation applied to $\mathbf{x}_0$. Our method of finding pertinent positives/negatives is formulated as an optimization problem over the perturbation variable $\boldsymbol{\delta}$ that is used to explain the model's prediction results. We denote the prediction of the model on the example $\mathbf{x}$ by $\text{Pred}(\mathbf{x})$, where $\text{Pred}(\cdot)$ is any function that outputs a vector of prediction scores for all classes, such as prediction probabilities and logits (unnormalized probabilities) that are widely used in neural networks, among others.

To ensure the modified example $\mathbf{x}$ is still close to the data manifold of natural examples, we propose to use an autoencoder to evaluate the closeness of $\mathbf{x}$ to the data manifold. We denote by $\text{AE}(\mathbf{x})$ the reconstructed example of $\mathbf{x}$ using the autoencoder $\text{AE}(\cdot)$.

### 3.1   Finding Pertinent Negatives (PN)

For pertinent negative analysis, one is interested in what is missing in the model prediction. For any natural example $\mathbf{x}_0$, we use the notation $\mathcal{X}/\mathbf{x}_0$ to denote the space of missing parts with respect to $\mathbf{x}_0$. We aim to find an interpretable perturbation $\boldsymbol{\delta} \in \mathcal{X}/\mathbf{x}_0$ to study the difference between the most probable class predictions in $\arg\max_i [\text{Pred}(\mathbf{x}_0)]_i$ and $\arg\max_i [\text{Pred}(\mathbf{x}_0 + \boldsymbol{\delta})]_i$. Given $(\mathbf{x}_0, t_0)$, our method finds a pertinent negative by solving the following optimization problem:

$$\min_{\boldsymbol{\delta} \in \mathcal{X}/\mathbf{x}_0} c \cdot f_\kappa^{\text{neg}}(\mathbf{x}_0, \boldsymbol{\delta}) + \beta\|\boldsymbol{\delta}\|_1 + \|\boldsymbol{\delta}\|_2^2 + \gamma\|\mathbf{x}_0 + \boldsymbol{\delta} - \text{AE}(\mathbf{x}_0 + \boldsymbol{\delta})\|_2^2. \tag{1}$$

We elaborate on the role of each term in the objective function (1) as follows. The first term $f_\kappa^{\text{neg}}(\mathbf{x}_0, \boldsymbol{\delta})$ is a designed loss function that encourages the modified example $\mathbf{x} = \mathbf{x}_0 + \boldsymbol{\delta}$ to be predicted as a different class than $t_0 = \arg\max_i [\text{Pred}(\mathbf{x}_0)]_i$. The loss function is defined as:

$$f_\kappa^{\text{neg}}(\mathbf{x}_0, \boldsymbol{\delta}) = \max\{[\text{Pred}(\mathbf{x}_0 + \boldsymbol{\delta})]_{t_0} - \max_{i \neq t_0}[\text{Pred}(\mathbf{x}_0 + \boldsymbol{\delta})]_i, -\kappa\} \tag{2}$$

where $[\text{Pred}(\mathbf{x}_0 + \boldsymbol{\delta})]_i$ is the $i$-th class prediction score of $\mathbf{x}_0 + \boldsymbol{\delta}$. The hinge-like loss function favors the modified example $\mathbf{x}$ to have a top-1 prediction class different from that of the original example $\mathbf{x}_0$. The parameter $\kappa \geq 0$ is a confidence parameter that controls the separation between $[\text{Pred}(\mathbf{x}_0 + \boldsymbol{\delta})]_{t_0}$ and $\max_{i \neq t_0}[\text{Pred}(\mathbf{x}_0 + \boldsymbol{\delta})]_i$. The second and the third terms $\beta\|\boldsymbol{\delta}\|_1 + \|\boldsymbol{\delta}\|_2^2$ in (1) are jointly called the elastic net regularizer, which is used for efficient feature selection in high-dimensional learning problems [44]. The last term $\|\mathbf{x}_0 + \boldsymbol{\delta} - \text{AE}(\mathbf{x}_0 + \boldsymbol{\delta})\|_2^2$ is an $L_2$ reconstruction error of $\mathbf{x}$ evaluated by the autoencoder. This is relevant provided that a well-trained autoencoder for the domain is obtainable. The parameters $c, \beta, \gamma, \geq 0$ are the associated regularization coefficients.

### 3.2   Finding Pertinent Positives (PP)

For pertinent positive analysis, we are interested in the critical features that are readily present in the input. Given a natural example $\mathbf{x}_0$, we denote the space of its existing components by $\mathcal{X} \cap \mathbf{x}_0$. Here we aim at finding an interpretable perturbation $\boldsymbol{\delta} \in \mathcal{X} \cap \mathbf{x}_0$ such that after removing it from $\mathbf{x}_0$, $\arg\max_i [\text{Pred}(\mathbf{x}_0)]_i = \arg\max_i [\text{Pred}(\boldsymbol{\delta})]_i$. That is, $\mathbf{x}_0$ and $\boldsymbol{\delta}$ will have the same top-1 prediction class $t_0$, indicating that the removed perturbation $\boldsymbol{\delta}$ is representative of the model prediction on $\mathbf{x}_0$. Similar to finding pertinent negatives, we formulate finding pertinent positives as the following optimization problem:

$$\min_{\boldsymbol{\delta} \in \mathcal{X} \cap \mathbf{x}_0} c \cdot f_\kappa^{\text{pos}}(\mathbf{x}_0, \boldsymbol{\delta}) + \beta\|\boldsymbol{\delta}\|_1 + \|\boldsymbol{\delta}\|_2^2 + \gamma\|\boldsymbol{\delta} - \text{AE}(\boldsymbol{\delta})\|_2^2, \tag{3}$$

**Algorithm 1** Contrastive Explanations Method (CEM)

---

**Input:** example $(x_0, t_0)$, neural network model $\mathcal{N}$ and (optionally ($\gamma > 0$)) an autoencoder $AE$
1) Solve (1) and obtain,
$\boldsymbol{\delta}^{\text{neg}} \leftarrow \operatorname{argmin}_{\boldsymbol{\delta} \in \mathcal{X}/\mathbf{x}_0} \ c \cdot f_\kappa^{\text{neg}}(\mathbf{x}_0, \boldsymbol{\delta}) + \beta\|\boldsymbol{\delta}\|_1 + \|\boldsymbol{\delta}\|_2^2 + \gamma\|\mathbf{x}_0 + \boldsymbol{\delta} - \text{AE}(\mathbf{x}_0 + \boldsymbol{\delta})\|_2^2$.
2) Solve (3) and obtain,
$\boldsymbol{\delta}^{\text{pos}} \leftarrow \operatorname{argmin}_{\boldsymbol{\delta} \in \mathcal{X} \cap \mathbf{x}_0} \ c \cdot f_\kappa^{\text{pos}}(\mathbf{x}_0, \boldsymbol{\delta}) + \beta\|\boldsymbol{\delta}\|_1 + \|\boldsymbol{\delta}\|_2^2 + \gamma\|\boldsymbol{\delta} - \text{AE}(\boldsymbol{\delta})\|_2^2$.
**return** $\boldsymbol{\delta}^{\text{pos}}$ and $\boldsymbol{\delta}^{\text{neg}}$. {Our Explanation: Input $x_0$ is classified as class $t_0$ because features $\boldsymbol{\delta}^{\text{pos}}$ are present and because features $\boldsymbol{\delta}^{\text{neg}}$ are absent. Code at `https://github.com/IBM/Contrastive-Explanation-Method` }

---

where the loss function $f_\kappa^{\text{pos}}(\mathbf{x}_0, \boldsymbol{\delta})$ is defined as

$$f_\kappa^{\text{pos}}(\mathbf{x}_0, \boldsymbol{\delta}) = \max\{\max_{i \neq t_0}[\text{Pred}(\boldsymbol{\delta})]_i - [\text{Pred}(\boldsymbol{\delta})]_{t_0}, -\kappa\}. \tag{4}$$

In other words, for any given confidence $\kappa \geq 0$, the loss function $f_\kappa^{\text{pos}}$ is minimized when $[\text{Pred}(\boldsymbol{\delta})]_{t_0}$ is greater than $\max_{i \neq t_0}[\text{Pred}(\boldsymbol{\delta})]_i$ by at least $\kappa$.

### 3.3 Algorithmic Details

We apply a projected fast iterative shrinkage-thresholding algorithm (FISTA) [2] to solve problems (1) and (3). FISTA is an efficient solver for optimization problems involving $L_1$ regularization. Take pertinent negative as an example, assume $\mathcal{X} = [-1, 1]^p$, $\mathcal{X}/\mathbf{x}_0 = [0, 1]^p$ and let $g(\boldsymbol{\delta}) = f_\kappa^{\text{neg}}(\mathbf{x}_0, \boldsymbol{\delta}) + \|\boldsymbol{\delta}\|_2^2 + \gamma\|\mathbf{x}_0 + \boldsymbol{\delta} - \text{AE}(\mathbf{x}_0 + \boldsymbol{\delta})\|_2^2$ denote the objective function of (1) without the $L_1$ regularization term. Given the initial iterate $\boldsymbol{\delta}^{(0)} = \mathbf{0}$, projected FISTA iteratively updates the perturbation $I$ times by

$$\boldsymbol{\delta}^{(k+1)} = \Pi_{[0,1]^p}\{S_\beta(\mathbf{y}^{(k)} - \alpha_k \nabla g(\mathbf{y}^{(k)}))\}; \tag{5}$$

$$\mathbf{y}^{(k+1)} = \Pi_{[0,1]^p}\{\boldsymbol{\delta}^{(k+1)} + \frac{k}{k+3}(\boldsymbol{\delta}^{(k+1)} - \boldsymbol{\delta}^{(k)})\}, \tag{6}$$

where $\Pi_{[0,1]^p}$ denotes the vector projection onto the set $\mathcal{X}/\mathbf{x}_0 = [0, 1]^p$, $\alpha_k$ is the step size, $\mathbf{y}^{(k)}$ is a slack variable accounting for momentum acceleration with $\mathbf{y}^{(0)} = \boldsymbol{\delta}^{(0)}$, and $S_\beta : \mathbb{R}^p \mapsto \mathbb{R}^p$ is an element-wise shrinkage-thresholding function defined as

$$[S_\beta(\mathbf{z})]_i = \begin{cases} \mathbf{z}_i - \beta, & \text{if } \mathbf{z}_i > \beta; \\ 0, & \text{if } |\mathbf{z}_i| \leq \beta; \\ \mathbf{z}_i + \beta, & \text{if } \mathbf{z}_i < -\beta, \end{cases} \tag{7}$$

for any $i \in \{1, \ldots, p\}$. The final perturbation $\boldsymbol{\delta}^{(k^*)}$ for pertinent negative analysis is selected from the set $\{\boldsymbol{\delta}^{(k)}\}_{k=1}^I$ such that $f_\kappa^{\text{neg}}(\mathbf{x}_0, \boldsymbol{\delta}^{(k^*)}) = 0$ and $k^* = \arg\min_{k \in \{1, \ldots, I\}} \beta\|\boldsymbol{\delta}\|_1 + \|\boldsymbol{\delta}\|_2^2$. A similar projected FISTA optimization approach is applied to pertinent positive analysis.

Eventually, as seen in Algorithm 1, we use both the pertinent negative $\boldsymbol{\delta}^{\text{neg}}$ and the pertinent positive $\boldsymbol{\delta}^{\text{pos}}$ obtained from our optimization methods to explain the model prediction. The last term in both (1) and (3) will be included only when an accurate autoencoder is available, else $\gamma$ is set to zero.

## 4 Experiments

This section provides experimental results on three representative datasets, including the handwritten digits dataset MNIST, a procurement fraud dataset obtained from a large corporation having millions of invoices and tens of thousands of vendors, and a brain imaging fMRI dataset containing brain activity patterns for both normal and autistic individuals. We compare our approach with previous state-of-the-art methods and demonstrate our superiority in being able to generate more accurate and intuitive explanations. Implementation details of projected FISTA are given in the supplement.

### 4.1 Handwritten Digits

We first report results on the handwritten digits MNIST dataset. In this case, we provide examples of explanations for our method with and without an autoencoder.

### 4.1.1 Setup

The handwritten digits are classified using a feed-forward convolutional neural network (CNN) trained on 60,000 training images from the MNIST benchmark dataset. The CNN has two sets of convolution-convolution-pooling layers, followed by three fully-connected layers. Further details about the CNN whose test accuracy was 99.4% and a detailed description of the CAE which consists of an encoder and a decoder component are given in the supplement.

### 4.1.2 Results

Our CEM method is applied to MNIST with a variety of examples illustrated in Figure 2. In addition to what was shown in Figure 1 in the introduction, results using a convolutional autoencoder (CAE) to learn the pertinent positives and negatives are displayed. While results without an CAE are quite convincing, the CAE clearly improves the pertinent positives and negatives in many cases. Regarding pertinent positives, the cyan highlighted pixels in the column with CAE (CAE CEM PP) are a superset to the cyan-highlighted pixels in column without (CEM PP). While these explanations are at the same level of confidence regarding the classifier, explanations using an AE are visually more interpretable. Take for instance the digit classified as a 2 in row 2. A small part of the tail of a 2 is used to explain the classifier without a CAE, while the explanation using a CAE has a much thicker tail and larger part of the vertical curve. In row 3, the explanation of the 3 is quite clear, but the CAE highlights the same explanation but much thicker with more pixels. The same pattern holds for pertinent negatives. The horizontal line in row 4 that makes a 4 into a 9 is much more pronounced when using a CAE. The change of a predicted 7 into a 9 in row 5 using a CAE is much more pronounced. The other rows exhibit similar patterns, and further examples can be found in the supplement.

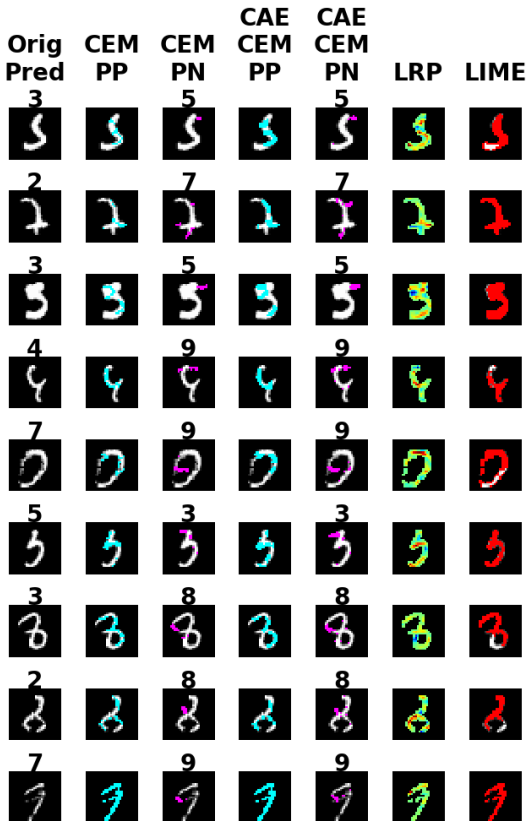

Figure 2: CEM versus LRP and LIME on MNIST. PP/PN are highlighted in cyan/pink respectively. For LRP, green is neutral, red/yellow is positive relevance, and blue is negative relevance. For LIME, red is positive relevance and white is neutral.

The two state-of-the-art methods we use for explaining the classifier in Figure 2 are LRP and LIME. LRP experiments used the toolbox from [21] and LIME code was adapted from https://github.com/marcotcr/lime. LRP has a visually appealing explanation at the pixel level. Most pixels are deemed irrelevant (green) to the classification (note the black background of LRP results was actually neutral). Positively relevant pixels (yellow/red) are mostly consistent with our pertinent positives, though the pertinent positives do highlight more pixels for easier visualization. The most obvious such examples are row 3 where the yellow in LRP outlines a similar 3 to the pertinent positive and row 6 where the yellow outlines most of what the pertinent positive provably deems necessary for the given prediction. There is little negative relevance in these examples, though we point out two interesting cases. In row 4, LRP shows that the little curve extending the upper left of the 4 slightly to the right has negative relevance (also shown by CEM as not being positively pertinent). Similarly, in row 3, the blue pixels in LRP are a part of the image that must obviously be deleted to see a clear 3. LIME is also visually appealing. However, the results are based on superpixels - the images were first segmented and relevant segments were discovered. This explains why most of the pixels forming the digits are found relevant. While both methods give important intuitions, neither illustrate what is necessary and sufficient about the classifier results as does our contrastive explanations method.

| ID | Risk | Events | PP | PN | Expert Feedback |
|----|------|--------|-----|-----|-----------------|
| 1 | Low | 1, 2, 9 | 2, 9 | 7 | ... vendor being registered and having a DUNs number makes the invoice low risk. However, if it came from a low CPI country then the risk would be uplifted given that the invoice amount is already high. |
| 2 | Medium | 2, 4, 7 | 2, 4 | 6 | ... the vendor being registered with the company keeps the risk manageable given that it is a risky commodity code. Nonetheless, if he was part of any of the FPL lists the invoice would most definitely be blocked. |
| 3 | High | 1, 4, 5, 11 | 1, 4, 11 | 2, 9 | ... the high invoice amount, the risky commodity code and no physical address makes this invoice high risk. The risk level would definitely have been somewhat lesser if the vendor was registered in VMF and DUNs. |

Table 2: Above we see 3 example invoices (IDs anonymized), one at low risk, one at medium and one at high risk level. The corresponding events that triggered and the PPs and PNs identified by our method are shown. We also report human expert feedback, which validates the quality of our explanations. The numbers that the events correspond to are given in Section 4.2.1.

## 4.2 Procurement Fraud

In this experiment, we evaluated our methods on a real procurement dataset obtained from a large corporation. This nicely complements our other experiments on image datasets.

### 4.2.1 Setup

The data spans a one-year period and consists of millions of invoices submitted by over tens of thousands vendors across 150 countries. The invoices were labeled as being either low risk, medium risk, or high risk based on a large team that approves these invoices. To make such an assessment, besides just the invoice data, we and the team had access to multiple public and private data sources such as vendor master file (VMF), risky vendors list (RVL), risky commodity list (RCL), financial index (FI), forbidden parties list (FPL) [4, 37], country perceptions index (CPI) [18], tax havens list (THL) and Dun & Bradstreet numbers (DUNs) [3]. Details describing each of these data sources are given in the supplement.

Based on the above data sources, there are tens of features and events whose occurrence hints at the riskiness of an invoice. Here are some representative ones. 1) if the spend with a particular vendor is significantly higher than with other vendors in the same country, 2) if a vendor is registered with a large corporation and thus its name appears in VMF, 3) if a vendor belongs to RVL, 4) if the commodity on the invoice belongs to RCL, 5) if the maturity based on FI is low, 6) if vendor belongs to FPL, 7) if a vendor

| Method | PP % Match | PN % Match |
|--------|-----------|-----------|
| CEM | **90.3** | **94.7** |
| LIME | 86.6 | N/A |
| LRP | 88.2 | N/A |

Table 1: Above we see the percentage of invoices on which the explanations were deemed acceptable by experts. For LIME and LRP we picked positively relevant features as proxies for PPs.

is in a high risk country (i.e. CPI < 25), 8) if a vendor or its bank account is located in a tax haven, 9) if a vendor has a DUNs number, 10) if a vendor and the employee bank account numbers match, 11) if a vendor only possesses a PO box with no street address.

With these data, we trained a three-layer neural network with fully connected layers, 512 rectified linear units and a three-way softmax function. The 10-fold cross validation accuracy of the network was high (91.6%).

### 4.2.2 Results

With the help of domain experts, we evaluated the different explanation methods. We randomly chose 15 invoices that were classified as low risk, 15 classified as medium risk and 15 classified as high risk. We asked for feedback on these 45 invoices in terms of whether or not the pertinent positives and pertinent negatives highlighted by each of the methods was suitable to produce the classification. To evaluate each method, we computed the percentage of invoices with explanations agreed by the experts based on this feedback.

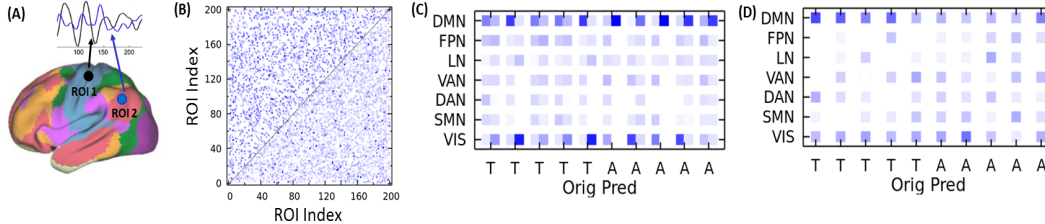

Figure 3: CEM versus LRP on pre-processed resting-state brain fMRI connectivity data from the open-access ABIDE I database. (A) Seven networks of functionally coupled regions across the cerebral cortex [8]. Color scheme: Purple: Visual (VIS), blue: Somatomotor (SMN), green: Dorsal Attention (DAN), violet: Ventral Attention (VAN), cream; Limbic (LN), orange: Frontoparietal (FPN), and red: default mode (DMN). (B) CEM PPs/PNs of a classified autistic brain are in the upper/lower triangle respectively. (C) A network-level view of the ROIs (region of interest) involving PP and PN functional connections (FCs) in the classified autistic (denoted as A) and neurotypical (denoted as T) subjects. For both (B) and (C), bolder the color higher the strength of the PP and PN FCs. (D) For LRP, positive relevance of FCs is depicted in a similar manner as in (C).

In Table 1, we see the percentage of times the pertinent positives matched with the experts judgment for the different methods as well as additionally the pertinent negatives for ours. We observe that in both cases our explanations closely match human judgment. We of course used proxies for the competing methods as neither of them identify PPs or PNs. There were no really good proxies for PNs as negatively relevant features are conceptually quite different as discussed in the supplement.

Table 2 shows 3 example invoices, one belonging to each class and the explanations produced by our method along with the expert feedback. We see that the expert feedback validates our explanations and showcases the power of pertinent negatives in making the explanations more complete as well as intuitive to reason with. An interesting aspect here is that the medium risk invoice could have been perturbed towards low risk or high risk. However, our method found that it is closer (minimum perturbation) to being high risk and thus suggested a pertinent negative that takes it into that class. *Such informed decisions can be made by our method as it searches for the most "crisp" explanation, arguably similar to those of humans.*

### 4.3 Brain Functional Imaging

In this experiment we look at explaining why a certain individual was classified as autistic as opposed to a normal/typical individual.

#### 4.3.1 Setup

The brain imaging dataset employed in this study is the Autism Brain Imaging Data Exchange (ABIDE) I [11], a large publicly available dataset consisting of resting-state fMRI acquisitions of subjects diagnosed with autism spectrum disorder (ASD), as well as of neuro-typical individuals. Precise details about standard ways in which this data was preprocessed is given in the supplement. Eventually, we had a 200x200 connectivity matrix consisting of real valued correlations for each subject. There were 147 ASD and 146 typical subjects.

We trained a single-layer neural network model on TensorFlow. The parameters of the model were regularized by an elastic-net regularizer. The leave-one-out cross validation testing accuracy is around 61.17% that matches the state-of-the-art results [28, 14, 38]. The logits of this network are used as model prediction scores, and we set $\mathcal{X} = [0,1]^p$, $\mathcal{X}/\mathbf{x}_0 = [0,1]^p/\mathbf{x}_0$ and $\mathcal{X} \cap \mathbf{x}_0 = [0,1]^p \cap \mathbf{x}_0$ for any natural example $\mathbf{x}_0 \in \mathcal{X}$.

#### 4.3.2 Results

With the help of domain experts, we evaluated the performance of CEM and LRP, which performed the best. LIME was challenging to use in this case, since the brain activity patterns are spread over the whole image and no reasonable segmentation of the images forming superpixels was achievable here. Per pixel regression results were significantly worse than LRP.

Ten subjects were randomly chosen, of which five were classified as autistic and the rest as neuro-typical. Since the resting-state functional connectivity within and between large-scale brain functional networks [42] (see Fig. 3A) are often found to be altered in brain disorders including autism, we decided to compare the performance of CEM and LRP in terms of identifying those atypical patterns. Fig. 3B shows the strong pertinent positive (upper triangle) and pertinent negative (lower triangle) functional connections (FC) of a classified ASD subject produced by the CEM method. We further group these connections with respect to the associated brain network (Fig. 3C). Interestingly, in four out of five classified autistic subjects, pertinent positive FCs are mostly (with a probability $> 0.26$) associated with the visual network (VIS, shown in purple in Fig 3A). On the other hand, pertinent negative FCs in all five subjects classified as autistic preferably (with a probability $> 0.42$) involve the default mode network (DMN, red regions in Fig. 3A). This trend appears to be reversed in subjects classified as typical (Fig. 3C). In all five typical subjects, pertinent positive FCs involve DMN (with probability $> 0.25$), while the pertinent negative FCs correspond to VIS. Taken together, these results are consistent with earlier studies, suggesting atypical pattern of brain connectivity in autism [16]. The results obtained using CEM further suggest under-connectivity in DMN and over-connectivity in visual network, in agreement with prior findings [16, 23]. LRP also identifies positively relevant FCs that mainly involve DMN regions in all five typical subjects (Fig. 3D). However, LRP associates positively relevant FCs from the visual network in only 40% of autistic subjects (Fig. 3D). These findings imply superior performance of CEM compared to LRP in robust identification of pertinent positive information from brain functional connectome data of different populations. The extraction of pertinent positive and negative features by CEM can further help reduce error (false positives and false negatives) in such diagnoses.

### 4.4 Quantitative Evaluation

In all the above experiments we also quantitatively evaluated our results by passing the PPs, and the PNs added to the original input, as independent inputs to the corresponding classifiers. We wanted to see here the percentage of times the PPs are classified into the same class as the original input and analogously the percentage of times the addition of PNs produced a different classification than the original input. This type of quantitative evaluation is similar to previous studies [32].

We found for both these cases and on all three datasets that our PPs and PNs are 100% effective in maintaining or switching classes respectively. This means that our approach can be trusted in producing highly informative and potentially sparse (or minimal) PPs and PNs that are also predictive on diverse domains.

## 5 Discussion

In the previous sections, we showed how our method can be effectively used to create meaningful explanations in different domains that are presumably easier to consume as well as more accurate. It's interesting that pertinent negatives play an essential role in many domains, where explanations are important. As such, it seems though that they are most useful when inputs in different classes are "close" to each other. For instance, they are more important when distinguishing a diagnosis of flu or pneumonia, rather than say a microwave from an airplane. If the inputs are extremely different then probably pertinent positives are sufficient to characterize the input, as there are likely to be many pertinent negatives, which will presumably overwhelm the user.

We believe that our explanation method CEM can be useful for other applications where the end goal may not be to just obtain explanations. For instance, we could use it to choose between models that have the same test accuracy. A model with possibly better explanations may be more robust. We could also use our method for model debugging, i.e., finding biases in the model in terms of the type of errors it makes or even in extreme case for model improvement.

In summary, we have provided a novel explanation method called CEM, which finds not only what should be minimally present in the input to justify its classification by black box classifiers such as neural networks, but also finds contrastive perturbations, in particular, additions, that should be necessarily absent to justify the classification. To the best of our knowledge this is the first explanation method that achieves this goal. We have validated the efficacy of our approach on multiple datasets from different domains, and shown the power of such explanations in terms of matching human intuition, thus making for more complete and well-rounded explanations.

## Acknowledgement

We would like to thank the anonymous reviewers for their constructive comments.

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
