[Supplementary Material · explanations-based-missing-final-supp.pdf]

# Supplemental Material

This supplementary material contains additional details about experiments and results.

## A    Experiments: FISTA details

As to the implementation of the projected FISTA for finding pertinent negatives and pertinent positives, we set the regularization coefficients $\beta = 0.1$, and $\gamma = \{0, 100\}$. The parameter $c$ is set to 0.1 initially, and is searched for 9 times guided by run-time information. In each search, if $f_\kappa$ never reaches 0, then in the next search, $c$ is multiplied by 10, otherwise it is averaged with the current value for the next search. For each search in $c$, we run $I = 1000$ iterations using the SGD solver provided by TensorFlow. The initial learning rate is set to be 0.01 with a square-root decaying step size. The best perturbation among all searches is used as the pertinent positive/negative for the respective optimization problems.

## B    MNIST

### B.1    Setup

The handwritten digits are classified using a feed-forward convolutional neural network (CNN) trained on 60,000 training images from the MNIST benchmark dataset. The CNN has two sets of convolution-convolution-pooling layers, followed by three fully-connected layers. All the convolution layers use a ReLU activation function, while the pooling layers use a $2 \times 2$ max-pooling kernel to downsample each feature map from their previous layer. In the first set, both the convolution layers contain 32 filters, each using a $3 \times 3 \times D$ kernel, where $D$ is an appropriate kernel depth. Both the convolution layers in the second set, on the other hand, contain 64 filters, each again using a $3 \times 3 \times D$ kernel. The three fully-connected layers have 200, 200 and 10 neurons, respectively. The test accuracy of the CNN is around 99.4%. The logits of this CNN are used as model prediction scores, and we set $\mathcal{X} = [-0.5, 0.5]^p$, $\mathcal{X}/\mathbf{x}_0 = [0, 0.5]^p/\mathbf{x}_0$ and $\mathcal{X} \cap \mathbf{x}_0 = [0, 0.5]^p \cap \mathbf{x}_0$ for any natural example $\mathbf{x}_0 \in \mathcal{X}$.

The CAE architecture contains two major components: an encoder and a decoder. The encoder compresses the $28 \times 28$ input image down to a $14 \times 14$ feature map using the architecture of convolution-convolution-pooling-convolution. Both of the first two convolution layers contain 16 filters, each using a $3 \times 3 \times D$ kernel, where $D$ is again an appropriate kernel depth. They also incorporate a ReLU activation function in them. The pooling layer is of the max-pooling type with a $2 \times 2$ kernel. The last convolution layer has no activation function, but instead has a single filter with a $3 \times 3 \times D$ kernel. The decoder, on the other hand, recovers an image of the original size from the feature map in the latent space. It has an architecture of convolution-upsampling-convolution-convolution. Again, both of the first two convolution layers have a ReLU activation function applied to the outputs of the 16 filters, each with a $3 \times 3 \times D$ kernel. The upsampling layer enlarges its input feature maps by doubling their side length through repeating each pixel four times. The last convolution layers has a single filter with the kernel size $3 \times 3 \times D$.

### B.2    Additional MNIST Results

Our CEM method is applied to MNIST on more examples and the results are illustrated in Figure 4. Regarding pertinent positives, again it can be seen that explanations using a CAE are visually more interpretable. In the first row, the outline of the loop of a 6 is more pronounced by the CAE. In the second row, the almost complete outline of the 5 is also much more pronounced when using a CAE, and similarly in the third row regarding the 3. Again, the same trend holds for pertinent negatives. In the first row, a few extra pixels are used to transform the 6 to a 4 and clearly make the transformation more explicit. In the fourth row, the loop that turns a 1 into a 6 is much thicker when using a CAE. The CAE in the fifth row does a much better job at lining up the extra pixels to turn a 1 into a 7. Transformation of the 0 to an 8 in the sixth row is particularly interesting. The bottom and top loops should have similar hole sizes, which is enforced better by the CAE with additional pixels added to the bottom loop.

Figure 4: CEM versus LRP and LIME on MNIST. PP/PN are highlighted in cyan/pink respectively. For LRP, green is neutral, red/yellow is positive relevance, and blue is negative relevance. For LIME, red is positive relevance and white is neutral.

## C  Procurement Fraud: Dataset Details

The VMF has information such as names of the vendors registered with the company, their addresses, account numbers and date of registration. The RVL and RCL contain lists of potentially fraudulent vendors and commodities that are often easy to manipulate. The FI contains information such as maturity of a vendor and their stock trends. The FPL released by the US government every year has two lists of suspect businesses. The CPI is a public source scoring (0-100) the risk of doing business in a particular country. The lower the CPI for a country, the worse the perception and hence higher the risk. Tax havens are countries such as the Cayman Islands where the taxes are minimal and complete privacy is maintained regarding people's financials. Dun & Bradstreet offers a unique DUNS number and DUNS name for each business registered with them. A DUNS ID provides a certain level of authenticity to the business.

## D  Brain Functional Imaging: Dataset Details

The brain imaging dataset employed in this study is the Autism Brain Imaging Data Exchange (ABIDE) I [11], a large publicly available dataset consisting of resting-state fMRI acquisitions of subjects diagnosed with autism spectrum disorder (ASD), as well as of neuro-typical individuals. Resting state fMRI provides neural measurements of the functional relationship between brain regions and is particularly useful for investigating clinical populations. Previously preprocessed acquisitions were downloaded (http://preprocessedconnectomes-project.org/abide/). We used the C-PAC preprocessing pipeline which included slice-time correction, motion correction, skull-stripping, and nuisance signal regression. Functional data was band-pass filtered (0.01—0.1 Hz) and spatially registered using a nonlinear method to a template space (MNI152). We limited ourselves to acquisitions with repetition time of 2s (sites NYU, SDSU, UM, USM) that were included in the original study of Di Martino et al. [11] and that passed additional manual quality control, resulting in a total of 147 ASD and 146 typical subjects (right-handed male, average age 16.5 yr). The CC200 functional parcellation atlas [8] of the brain, totaling 200 regions, was used to estimate the brain connectivity matrix. The mean time series

for regions of interest (ROI) was extracted for each subject. A Pearson product-moment correlation was calculated for the average of the time series of the ROI (see Fig. 3A) to build a 200x200 connectivity matrix for each subject. Only positive correlation values in functional connectivity matrices were considered in this study.