[Reviews · NeurIPS 2018]

Reviewer 1



This paper proposes contrastive explanations to explain predictions from machine learning models. The idea is very elegant: this paper observes that most current work in interpretable machine learning only use features that are present in the instance, but human reasoning also relies on minimal differences from the closest class. In particular, I like the argument based on the notion of pertinent positives and pertinent negatives in medicine and criminology. The mathematical formulation of pertinent negatives is close to perturbation analysis and it is clever to use autoencoders to make the generated instance close to real examples. But the formulation of the pertinent positives may require a bit of more discussion. It seems straightforward and an intuitive adaptation of the pertinent negatives. But there is no requirement that the chosen parts are meaningful. This is indeed be a hard problem, but this may be related to the relative worse performance of the pertinent positives in the experiments too. It seems that the pertinent positives are much harder to interpret, at least for me. Another implicit requirement in the proposed approach is that x_0 falls in discrete space, which may be an issue in some settings. The main weakness of this paper lies in the evaluation. Although it is a great thing that this paper uses more datasets than MNIST, the evaluation can be much improved. 1) The statements in the MNIST experiment such as "While results without an CAE are quite convincing, the CAE clearly improves the pertinent positives and negatives in many cases. Regarding pertinent positives, the cyan highlighted pixels in the column with CAE (CAE CEM PP) are a superset to the cyan-highlighted pixels in column without (CEM PP). While these explanations are at the same level of confidence regarding the classifier, explanations using an AE are visually more interpretable." are problematic. These are quite subjective statements, and some form of quantitative evaluation across subjects is required for such claims. 2) In the procurement fraud experiment, it seems that the experts like everything that the algorithm shows. Risk evaluation seems a non-trivial problem. It is unclear whether these experts or humans are good at this task. Also, given the sample size, it is unclear whether the difference in Table 1 is statistically significant. 3) This paper did not provide enough information regarding how the evaluation was done in the brain functional imaging experiment. It seems that the only sentence is "With the help of domain experts". 4) c, \beta, and \gamma are important parameters for the proposed approach. The main paper did not discuss the choice of these parameters at all, and the supplementary material only gives procedural information. It would be great if this paper provides more thoughtful discussions on the choices of these parameters, or maybe the insensitivity of these parameters if that is the case. Overall, I really like the idea of this paper and believe that this paper should be accepted. Given the space limit of NIPS submissions, one possible way to improve the paper is to drop one experiment and make the other two experiments more solid. Minor presentation-related suggestions: I like the introduction overall, but the first sentence seems a bit out of nowhere and statements such as "Explanations as such are used frequently by people" are questionable and at least requires better evidence. line 218: an CAE -> a CAE line 252: spend -> spending I have read the review and it would be useful if the user can clarify how some set operations in the formulation apply to continuous variables.

Reviewer 2



The paper proposes a Contrastive Explanation Method (CEM) with the goal of justifying, or explaining, the decisions made by deep neural networks. A notable characteristic of the proposed method is that for every class of interest it aims at identifying two sets of features that can be used to jutify the decisions related to a given class. The first type, pertinent positives (PN), are features whose occurrence are the minimum necessary to justify the classification prediction. The second type, pertinent negatives (PN) are features whose absence is necessary to justify the prediction of a given class. Given an input image, explanations are given by extending the output by highlighting these two types of features. Experiments on the MNIST [38], ABIDE-I [11] and a preocurement fraud dataset [9] show the performance of the proposed method. =================== Overall the content of the manuscript is sound, clear and easy to follow. I find valuable the fact that the evaluation includes experiments datasets on multiples data modalities, i.e. visual data (Sec. 4.1 and 4.3) and text-based data (Sec. 4.2). In addition, this seems to be one of the first works focusing explictly on the identification of pertinent negatives. In addition, there seems to be code available in the supplementary material. If the plan is to release code to run the proposed method, this could be good for reproducibility of the presented results. Having said this, my concerns with the manuscript are the following: Even when I find novel the fact that the proposed method is explictly designed to identify and indicate PN features, this is not the first work to highlight this type of negative feature. Examples of existing works include Samek et al., 2017 and more recently, Oramas et al. arXiv:1712.06302, which as part of their generated output provided some explanations supporting the absence of specific features. I consider positioning with respect to these works a must in order to effectively highlight the novelty brought by the proposed method. In Section 3, it is stated that the proposed method aims at identify the features that could serve as PPs and PNs for model explanation. However, in several of the figures/Tables displaying the explanations of the proposed method just one output of each type of feature (PP or PN) is given. To these outputs already highlight more than one feature of each type? if yes, how can we distinguish between the different features highlighted within a type. if not, an multiple outputs are posible, what is the criterion to select which features to highlight as part of the output? In Section 3 it is mentioned the need for an autoencoder (AE). DUring the evaluation, this autoencoder is only employed in 2/3 of the conducted experiments. At this point it is not clear to me when this autoencoder is necessary or not. Is there a principled means to verify whether this autoencoder is necessary? Experiments are conducted on relatively simple and small datasets. Moreover, the Procurement Fraud dataset used in Section 4.2 does not seem to be public. If this is indeed the case, this would limit future comparisons with respect to this work. In addition, while the manuscript stresses in the abstract the black box characteristic of deep neural networks (DNNs), the tested architectures are relatively simple and shallow. Therefore the potential of the proposed method is unclear in more complex scenarios. For the case of DNNs for visual data, I would suggest evaluating on one of the standard architectures, e.g. alexnet, VGG, ResNet, etc., for which pre-trained models can be found online. Results on one of this deeper more complex settings will strengthen the potential of the proposed method. Finally, the presented results are mostly qualitative. In this regard I would suggest performing a either: a) an occlussion analysis(Zeiler et al., ECCV 2014, Samek et al., 2017) , b) a pointing experiment (Zhang et al., ECCV 2016) or, c) a measurement of explanation accuracy by feature coverage (Oramas et al. arXiv:1712.06302). Any of these protocols should allow to quantitatively evaluate the relevance of the features highlighted by the proposed as part of the explanation. In addition, I find the set of provided qualitative examples quite reduced. In this regard, I encourage the authors to update the supplementary material in order to show extended qualitative results of the explanations produced by their method. I would appreciate if my concerns are addressed in the rebuttal. ========================== Full references ========================== Jose Oramas, Kaili Wang, Tinne Tuytelaars "Visual Explanation by Interpretation: Improving Visual Feedback Capabilities of Deep Neural Networks" (2017), arXiv:1712.06302. Wojciech Samek, Alexander Binder, Grégoire Montavon, Sebastian Lapuschkin, and Klaus-Robert Müller: "Evaluating the Visualization of What a Deep Neural Network has Learned" IEEE Transactions on Neural Networks and Learning Systems, 28(11):2660-2673, 2017 Matthew D. Zeiler and Rob Fergus. "Visualizing and understanding convolutional networks" ECCV, 2014. Jianming Zhang, Zhe Lin, Jonathan Brandt, Xiaohui Shen, Stan Sclaroff "Top-down Neural Attention by Excitation Backprop" ECCV 2016. ======================================== Post Rebuttal Update ======================================== I thank the authors for their rebuttal. The provided rebuttal satisfied to some extent most of the points I raised on my review. I share the R1's view that the proposed idea is interesting. Though, I do not consider the fact of highlighting features that will produce a negative effect that novel. Indeed, most current work in interpretable machine learning (e.g. Selvaraju et al., arXiv:1610.02391, Chattopadhay et al., WACV'18, Reddy et al., arXiv:1708.06670 ) only highlights features that are present in the instance. However, highlighting other, potentially conflicting, features related to other classes will mostly require generating such visualizations for other non-max classes with high score for the instance being explained. I share the concerns of R1 on that the evaluation aspect could be improved. Currently, results are reported mostly on a qualitative fashion. While some deterministic behavior is expected when giving the PPs and PNs as part of the input to the neural network, the identification of these PPs and PNs is not perfect. I still would like to see empirical (quantitative) evidence of to what these elements satisfy that expected behavior. Moreover, for the case of the MNIST experiment, the rebuttal (responding to R1) states that there is no widely accepted quantitative analysis for measuring explanations on visual datasets. Indeed, there is no unique metric for this yet. However, as I stated on my review, there are several evaluation methods that have been proposed and can help position the proposed method appropriately. At this point, the rebuttal does not change my initial opinion of the manuscript. I still consider the manuscript a good paper and a good match for NIPS.

Reviewer 3



The main contribution of this paper is the idea of a pertinent negative explanation, i.e. the idea of some features that, were they present in the instance, would change the classification. While not entirely new, it is presented here in a logical and convincing fashion. While the paper illustrates this extensively on MNIST (a pertinent positive if ever there was one!), the paper goes beyond this and experimentally evaluates it on two other, quite different domains.